# Theoretical framework for confined ion transport in two-dimensional nanochannels

Shouwei Liao[1], Yanchang Liu[1], Libo Li [1] ✉, Li Ding [2], Yanying Wei [1,3] ✉ & Haihui Wang [2] ✉

Quantitative understanding of ion transport mechanism is crucial for numerous applications of two-dimensional (2D) nanochannels, but is far from being resolved. Here, we formulated a theoretical framework for both self-diffusion and electromigration of hydrated monatomic ions in various 2D nanochannels (e.g. graphene, h-BN, g-$C_3N_4$, $MoS_2$), by molecular dynamics simulations. The self-diffusivity and mobility of ions in 2D nanochannels both increases linearly with ion-wall distance for small hydrated ions, yet keeps constant for large ones. The underlying mechanism reveals that when ions approach water-layers in nanochannels or possess large hydration shell, their hydration shells become severely distorted. This increases the free energy difference between hydration shell and the surrounding water-layers, water residence time in hydration shell and ion-water friction. Several involving quantitative relations were revealed, with Nernst–Einstein relation validated with both simulations and theoretical derivation. This work shows profound implications for various applications, including ion-sieving, nanodevices and nano-power generators, etc.

The ion transport through two-dimensional (2D) nanochannels[1–4] formed by graphene nanosheets[5–17] and other 2D materials[15,18–23] possesses broad application prospects in seawater desalination[12,14,24–27], osmotic power generation[18,28–30], nanodevices[5–7,19], etc. since transport through 2D nanochannels shows numerous unexpected phenomena distinct from transport in bulk solution such as memristor effect[5,6], transistor-like gating effect[19], complete steric exclusion[9,16]. These exceptional phenomena root from the unique ion transport mechanism in confined nanochannels which is mainly governed by two characteristics, (1) water molecules form layered structure in nanochannels[8,31,32] instead of uniform continuum of bulk water, (2) ion's hydration shell (HS) becomes distorted or even partially dehydrated[7,15,33]. Very recently, these unique water structures have been explored to regulate the ion transport dramatically. For instance, the position of an ion relative to water layers can affect the transport of hydrated ions[15]; the spatial and temporal correlations between ions

and water layers can lead to an ionic current with features of ionic rectifiers and logical gates[34]; the mobility of ions in nanochannels can depend on the polarization of water molecules[11] or the hydration strength of the ion[35]. However, most of current work only provides qualitative descriptions of these phenomena, rarely digging into the quantitative mechanisms. Though a recent article reports the correlation that the relative ion mobility ~ ion core diameter[15], the prediction error is ~75% and, therefore, not sufficient to yield more in-depth physical insights. Another crucial issue for these studies is that, the Nernst-Einstein relation (ion mobility $\mu \propto$ ion diffusivity $D$) is usually taken for granted, yet has seldom (if ever) been validated for nanochannels[15,30]. To validate such relation in 2D nanochannels is pivotal, as it was recently shown to breakdown in some 1D nanochannels[36]. Although how bulk continuum water influence the ion transport has recently been elucidated after a century of epics exploration by the science community[37], the physical nature of how

[1]State Key Laboratory of Pulp and Paper Engineering, School of Chemistry & Chemical Engineering, Guangdong Provincial Key Lab of Green Chemical Product Technology, South China University of Technology, Guangzhou, China. [2]Beijing Key Laboratory of Membrane Materials and Engineering, Department of Chemical Engineering, Tsinghua University, Beijing, China. [3]Quzhou Membrane Material Innovation Institute, Quzhou, China. ✉e-mail: celbli@scut.edu.cn; ceyywei@scut.edu.cn; cehhwang@tsinghua.edu.cn

the water layers and distorted HSs in 2D nanochannels regulate the ion transport remains far from being resolved. Not to mention further developing a mechanism to quantitatively predict the transport behavior, such as diffusion and electromigration, for various ions in nanochannels.

Here, the transport behavior of monatomic ions (Li$^+$, Na$^+$, K$^+$, Rb$^+$, Cs$^+$, Ca$^{2+}$, Mg$^{2+}$ and Cl$^-$) in various 2D nanochannels formed by assembling the nanosheets of graphene, h-BN, MoS$_2$, g-C$_3$N$_4$, involving ion self-diffusion and electromigration, are studied by molecular dynamics (MD) simulations. Several different force fields (FF)[38] are employed, such as OPLS-AA[39], Merz[40,41], Netz[42,43] and Williams FFs[44], the latter 3 are named by the authors who developed them. To describe ion-graphene interaction, two versions of Lennard-Jones (LJ) interaction parameters between ions and channel wall atoms ($\varepsilon_{I-W}$) were employed for each FF: one calculated by the Lorentz-Berthelot (LB) mixing rule (denoted as $\varepsilon_{I-W}^{LB}$) and the other recently optimized one which describes the ion-graphene interactions in solution accurately[44,45] (denoted as $\varepsilon_{I-W}^{ion-\pi}$, see Method section and Supplementary Tables 1, 2 for details). These extensive simulations reveal a concise rule: The ratio of the ion self-diffusivity in 2D nanochannels to that in bulk water ($D_{channel}/D_{bulk}$) for

ions with small radius of the 1$^{st}$ HS ($r_{HS}$) correlates linearly with the ion-water layer distance, while it keeps constant for the ions with large $r_{HS}$. The mechanism of how water layers and HSs of ions regulate $D_{channel}$ is in-depth elucidated on the basis of fundamental physics such as ion-water friction, the residence time ($\tau$) of water molecule in ion's HS, and the free energy profiles of water molecules around the ion. Quantitative relationships among these physical quantities are elucidated with physical implications well explained. Furthermore, the above mechanism is also proved to be valid for ion electromigration in 2D nanochannels, i.e. the Nernst-Einstein relation is thoroughly validated for the first time in 2D nanochannels with both simulation data and theoretical derivations.

## Results

### Position-dependent relative ion diffusivity

Among all simulations employing different FFs (Supplementary Tables 1) in graphene nanochannels, when the ion distribution profile changes (Fig. 1a, Supplementary Fig. 1), so does its $D_{channel}/D_{bulk}$ ratio (Fig. 1b, Supplementary Tables 2, 3). This finding inspired us to calculate the average ion-wall distance ($d_{ion-wall}$) with Eq. (1) (See

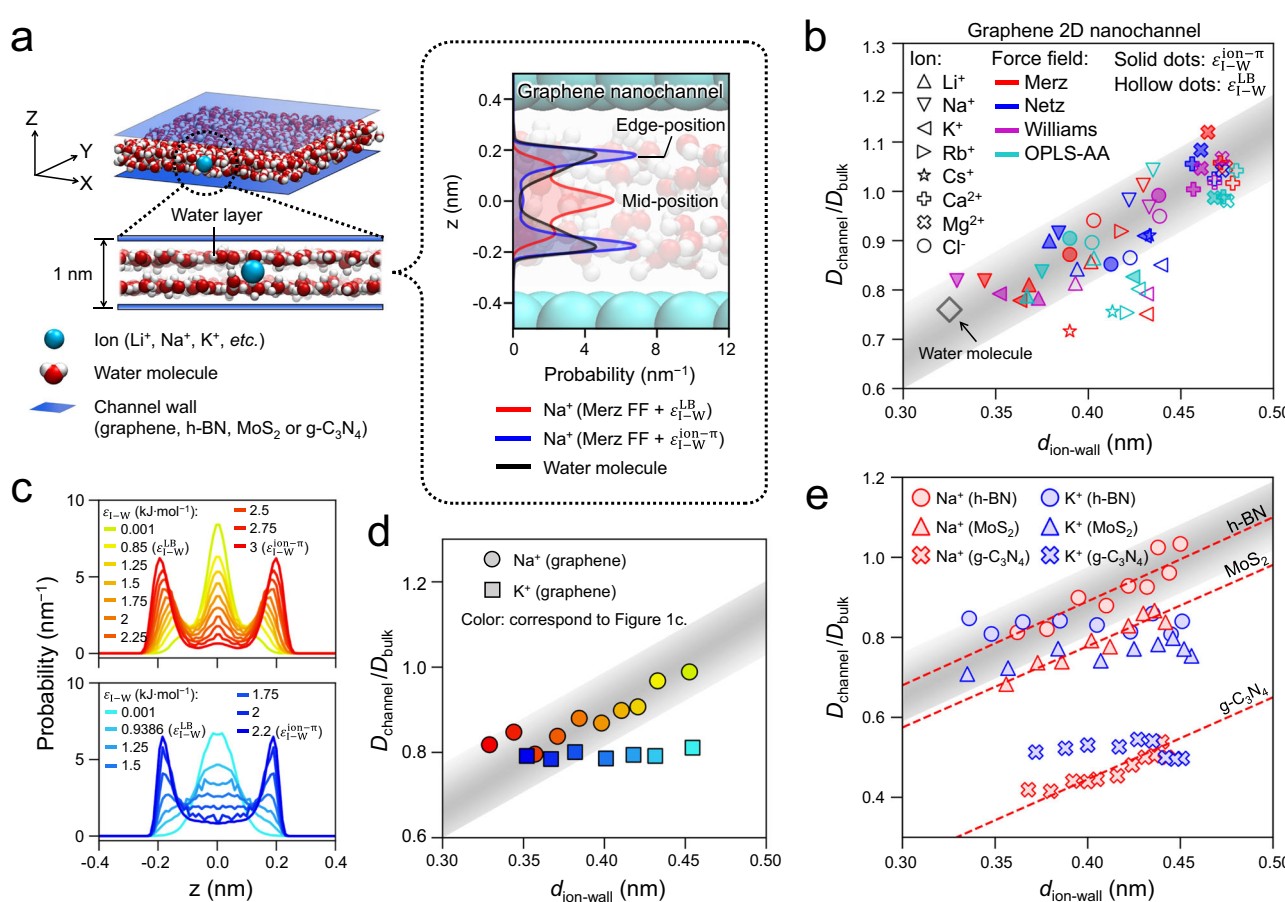

**Fig. 1 | Ratio of ion self-diffusivity ($D_{channel}/D_{bulk}$) increases linearly with increasing ion-wall distance ($d_{ion-wall}$) for hydrated ions with small $r_{HS}$, while it keeps constant for ions with large $r_{HS}$. a** Simulation system of hydrated ions in 2D nanochannels, with distribution profiles of Na$^+$ and water molecules in graphene 2D nanochannel. Distribution profiles were taken from simulations using Merz force field (FF)[40,41]. $\varepsilon_{I-W}^{LB}$ and $\varepsilon_{I-W}^{ion-\pi}$ represent the original and optimized[44,45] versions of the Lennard-Jones (LJ) parameters between ion and graphene, respectively (Supplementary Table 2). **b** $D_{channel}/D_{bulk}$ increases linearly with $d_{ion-wall}$ for all ions and FFs under study (shadowed area represents the prediction error), except ions with large $r_{HS}$ (e.g. K$^+$, Rb$^+$, Cs$^+$). Merz[40,41], Netz[42,43] and Williams[44] FFs (named by the authors who developed them) and OPLS-AA[39] FF were employed. The data for water

molecules (black hollow diamond) is shown for comparison. **c** Distribution profiles of Na$^+$ and K$^+$ in graphene 2D nanochannel simulated with Williams FF. The $\varepsilon_{I-W}$ (LJ parameters between ion and wall atoms) of Na$^+$ and K$^+$ were adjusted smoothly from 0.001 kJ·mol$^{-1}$ to corresponding $\varepsilon_{I-W}^{ion-\pi}$ values to gradually change their distribution profiles (**c**) and $d_{ion-wall}$ (**d**). Each color in (**c, d**) represents an employed $\varepsilon_{I-W}$. **d, e** $D_{channel}/D_{bulk}$ increases linearly with $d_{ion-wall}$ for Na$^+$ with small $r_{HS}$ while keeps constant for K$^+$ with large $r_{HS}$. This finding holds true for 2D nanochannels constructed by graphene (**d**), h-BN, MoS$_2$ or g-C$_3$N$_4$ nanosheets (**e**, $\varepsilon_{I-W}$ varies between 0.01 and 4 kJ·mol$^{-1}$, using Williams FF), also true for other studied ions (see Supplementary Fig. 8 for Li$^+$, Rb$^+$, Cs$^+$). The shadowed area in (**b, d, e**) are identical. Source data are provided as a Source Data file.

"Methods" section), which gives the following concise correlation: $D_{\text{channel}}/D_{\text{bulk}}$ linearly increases with increasing $d_{\text{ion-wall}}$ (Fig. 1b). However, the bulky alkali metal ions with large $r_{\text{HS}}$ ($\geq r_{\text{HS}}$ of K$^+$, see Supplementary Table 2 and Supplementary Fig. 2–5) usually give $D_{\text{channel}}/D_{\text{bulk}}$ lower than the linear correlation predicts for large $d_{\text{ion-wall}}$ position (Fig. 1b). Moreover, various 2D nanochannels assembled by nanosheets of different materials, such as graphene, h-BN, MoS$_2$, g-C$_3$N$_4$ were also simulated (Supplementary Fig. 6). In these simulations, LJ parameters between ion and channel wall atoms ($\varepsilon_{\text{I-W}}$) were adjusted to gradually change $d_{\text{ion-wall}}$ (Fig. 1c–e, Supplementary Fig. 7), i.e. with $\varepsilon_{\text{I-W}}$ increasing, the peaks of ion's distribution profile move closer to the channel walls (Fig. 1c), which reduces $d_{\text{ion-wall}}$. All these simulations confirm that $D_{\text{channel}}/D_{\text{bulk}}$ varies linearly with $d_{\text{ion-wall}}$ for ions with small $r_{\text{HS}}$, while it is constant and independent on $d_{\text{ion-wall}}$ for ions with large $r_{\text{HS}}$, like K$^+$ (Fig. 1d, e) in all kinds of studied 2D nanochannels, thereby we will focus on results of graphene 2D nanochannels hereafter. Furthermore, such $D_{\text{channel}}/D_{\text{bulk}}$ ~ $d_{\text{ion-wall}}$ correlation (varying linearly for Li$^+$ and Na$^+$; constant for bulky K$^+$, Rb$^+$ and Cs$^+$) holds true even when $d_{\text{ion-wall}}$ of the studied ions in graphene 2D nanochannels is controlled by restraining their z coordinates with harmonic potential (Supplementary Fig. 8, note that all other simulations in this work were performed without any artificial restrains). The linear correlation agrees well with very recent experiments on graphene 2D nanochannels[15], yet our results are more precise with a much smaller prediction error, e.g. 0.16 versus the reported ~0.75[15]. It can be attributed to that we simulated diluted solutions to avoid the interference of ion-ion interaction, and $D_{\text{channel}}/D_{\text{bulk}}$ correlates with $d_{\text{ion-wall}}$ instead of the reported ion-core diameter or near-wall probability[15]. More importantly, we validated such rule for divalent ions, and for many other 2D nanochannels besides graphene. Quite a few new fundamental insights emerge from our work for the first time, e.g. $D_{\text{channel}}/D_{\text{bulk}}$ of bulky ions (K$^+$, Rb$^+$ and Cs$^+$, see Fig. 1d, e and Supplementary Fig. 8) is independent on $d_{\text{ion-wall}}$; and $D_{\text{channel}}/D_{\text{bulk}}$ can be even above 1 for small ions with sufficiently large $d_{\text{ion-wall}}$ (Na$^+$, Ca$^{2+}$, Mg$^{2+}$ in Fig. 1b and Supplementary Fig. 8). In addition, though we define bulky and small ions based on $r_{\text{HS}}$, discussions with ionic radii[46] would yield similar results, due to the correlation between these radii (Supplementary Fig. 9). Note we usually discuss the $D_{\text{channel}}/D_{\text{bulk}}$ ratio instead of $D_{\text{channel}}$ (Supplementary Note 1 and Supplementary Fig. 10) to focus on a universal rule, i.e. how solvent structure (water layers) and ion motion in nanochannels differ from bulk solutions, instead of on specific details of individual ions or FFs.

For a 2D nanochannel accommodating one hydrated ion and water layers, only ion-wall and ion-water layer interactions can affect ion diffusion. Further simulations indicate that hydrated ion-wall interaction hardly affect $D_{\text{channel}}/D_{\text{bulk}}$, even when $\varepsilon_{\text{I-W}}$ parameters vary by two orders of magnitude (Supplementary Note 2 and Supplementary Fig. 11). Thus the above $D_{\text{channel}}/D_{\text{bulk}}$ ~ $d_{\text{ion-wall}}$ rule can mainly be attributed to the hydrated ion-water friction force as follows. When ions approach water layers (small $d_{\text{ion-wall}}$) or possess large $r_{\text{HS}}$ like K$^+$, they will contact closely with the water layers (Fig. 1 and Supplementary Fig. 8), i.e. the water density in the diffusion directions increases (Supplementary Fig. 12). Thus ions suffer from larger ion-water friction, and $D_{\text{channel}}$ decreases. Contrarily, ions possessing small $r_{\text{HS}}$ and large $d_{\text{ion-wall}}$ suffer from small friction, and their $D_{\text{channel}}/D_{\text{bulk}}$ could be even above 1 (Ca$^{2+}$ and Mg$^{2+}$ in Fig. 1b). However, two questions remain open: (i) What is the physical nature of ion-water friction in the 2D nanochannel, and can it be explained from fundamental physics such as force, energy or even HS structure? (ii) Can a mechanism be quantitatively developed? For instance, why is $D_{\text{channel}}/D_{\text{bulk}}$ of hydrated K$^+$ independent on $d_{\text{ion-wall}}$? The following discussions will focus on graphene 2D nanochannels to answer these questions in-depth, yet can also apply for other 2D nanochannels.

## Correlations of physics quantities

The well-known Einstein equation relates the diffusivity to friction force[37,47–49], $D = \frac{k_{\text{B}}T}{\lambda} = \frac{\gamma(k_{\text{B}}T)^2}{I_{\text{FACF}}}$ ($\lambda = \frac{I_{\text{FACF}}}{\gamma k_{\text{B}}T}$ is the ion-water friction coefficient, $\gamma$ is the dimensionality (3 for bulk solution, 2 for 2D nanochannel), $I_{\text{FACF}} = \int_0^\infty \text{FACF}(t)dt$, and FACF(t) refers to the Force Auto-Correlation Function, see "Methods" section). Thus, FACFs of the studied ions were calculated to analyze the ion-water friction force, as shown in Supplementary Figs. 13–18, where $D^{\text{FACF}}$ agree with $D^{\text{MSD}}$ (Supplementary Fig. 13, the superscript FACF and MSD indicate the diffusivity are calculated from FACF and MSD, respectively) and $D_{\text{channel}}^{\text{FACF}}/D_{\text{bulk}}^{\text{FACF}}$ ~ $d_{\text{ion-wall}}$ follows a similar rule as $D_{\text{channel}}^{\text{MSD}}/D_{\text{bulk}}^{\text{MSD}}$ ~ $d_{\text{ion-wall}}$ (Supplementary Fig. 14). Ions can be located in three different solvation environments or positions in this study, (i) bulk water, (ii) mid-position in nanochannels (the middle position inside the 2D nanochannel, $d_{\text{ion-wall}} = 0.5$ nm, Fig. 1a) with minimum water density, (iii) edge-position in nanochannels (the peak positions of ion distribution profiles which are close to water layers, $d_{\text{ion-wall}} < 0.5$ nm, Fig. 1a). As for a given ion, its $I_{\text{FACF}}^{\text{edge}}$ ($I_{\text{FACF}}$ for the ion locating at the edge-position) is usually ~20% larger than $I_{\text{FACF}}^{\text{bulk}}$ ($I_{\text{FACF}}$ for the ion in bulk water). However, $I_{\text{FACF}}^{\text{mid}}$ ($I_{\text{FACF}}$ at mid-position) can be larger than, similar to or even smaller than $I_{\text{FACF}}^{\text{bulk}}$ (Fig. 2a). The transport behavior of three representative ions is characterized as follows, (i) for Li$^+$, $1/I_{\text{FACF}}^{\text{edge}} < 1/I_{\text{FACF}}^{\text{bulk}} \leq 1/I_{\text{FACF}}^{\text{mid}}$ with $D_{\text{edge}}/D_{\text{bulk}} < 1 \leq D_{\text{mid}}/D_{\text{bulk}}$; (ii) for Na$^+$, $1/I_{\text{FACF}}^{\text{edge}} < 1/I_{\text{FACF}}^{\text{mid}} \approx 1/I_{\text{FACF}}^{\text{bulk}}$ with $D_{\text{edge}}/D_{\text{bulk}} < D_{\text{mid}}/D_{\text{bulk}} \approx 1$; (iii) for K$^+$, $1/I_{\text{FACF}}^{\text{edge}} \approx 1/I_{\text{FACF}}^{\text{mid}} < 1/I_{\text{FACF}}^{\text{bulk}}$ with $D_{\text{edge}}/D_{\text{bulk}} \approx D_{\text{mid}}/D_{\text{bulk}} < 1$ (showing a constant $D_{\text{channel}}/D_{\text{bulk}}$, independent on $d_{\text{ion-wall}}$, Fig. 1d). The following discussion is focused on these three ions. Other ions' $I_{\text{FACF}}$ and $D_{\text{channel}}/D_{\text{bulk}}$ just show features similar to either of them (Supplementary Tables 4-7). Further, the ions' FACFs could be divided into the head part (violently oscillating for t <~ 0.3 ps) and the following tail part (smoothly decaying to 0, Fig. 2b), with corresponding integral as $I_{\text{head}} = \int_0^{t_b} \text{FACF}(t)dt$ and $I_{\text{tail}} = \int_{t_b}^\infty \text{FACF}(t)dt$, where $t_b$ is the boundary time separating these two parts (Supplementary Fig. 15–18). Note when a given ion is located in different solvation environments, $I_{\text{head}}$ changes very slightly, while the change of $I_{\text{FACF}}$ mainly comes from $I_{\text{tail}}$ (Fig. 2c, Supplementary Note 3).

It is known that $I_{\text{FACF}}$ is closely related to the speed of water molecules moving around the ion[37], we further calculated the residence time ($\tau$) of water molecules in the 1st HS of ions (abbreviated as HS). Figure 2d shows that $\Delta I_{\text{FACF}}$ ($I_{\text{FACF}}^{\text{channel}} - I_{\text{FACF}}^{\text{bulk}}$) correlates well with $\tau_{\text{channel}}/\tau_{\text{bulk}}$. Specifically, when an ion moves from bulk water into the nanochannel, and if $\tau$ rises sharply ($\tau_{\text{channel}}/\tau_{\text{bulk}} > 3$), $I_{\text{FACF}}$ also increases considerably ($\Delta I_{\text{FACF}} > 500\times10^{12}$ N$^2$·mol$^{-2}$·s), e.g. for Li$^+$ and Na$^+$ at edge-position, as well as K$^+$ at both mid- & edge-positions (Fig. 2d). On the contrary, smaller $\tau_{\text{channel}}/\tau_{\text{bulk}}$ ($\leq 2$) leads to $\Delta I_{\text{FACF}} \approx 0$ for Na$^+$ at mid-position, or even below 0 for Li$^+$ at mid-position (Fig. 2d). Shorter $\tau$ indicates more frequent exchanges of water molecules between HS and the surrounding solvent, and a more rapid change of water configurations around the ion (Fig. 2e). Consequently, the correlation between water configurations around the ion with large time interval ($\Delta t > t_b$) becomes weaker, and FACF($\Delta t$) becomes closer to 0. Thus FACF's tail decays to 0 more quickly and $I_{\text{tail}}$ decreases, while $I_{\text{head}}$ hardly changes, leading to decreased $I_{\text{FACF}}$ consequently. Contrarily, longer $\tau$ indicates a slower change in the water configurations around the ion, and consequently a larger $I_{\text{tail}}$ and $I_{\text{FACF}}$.

Further, the ratio of $\tau_{\text{channel}}/\tau_{\text{bulk}}$ for the studied ions correlates with the shape of their HSs: Ions with small $\tau_{\text{channel}}/\tau_{\text{bulk}}$ (e.g. Li$^+$ or Na$^+$ at mid-position in nano-channel) show spherical HSs similar to those in bulk water (Fig. 3a). Contrarily, ions with large $\tau_{\text{channel}}/\tau_{\text{bulk}}$ (e.g. Li$^+$, Na$^+$ at edge-position in channel and K$^+$ at mid- or edge-position) show HSs distorted to rings & poles (Fig. 3a), due to the nanoconfinement effect overwhelming the z (channel height, or d-spacing direction)

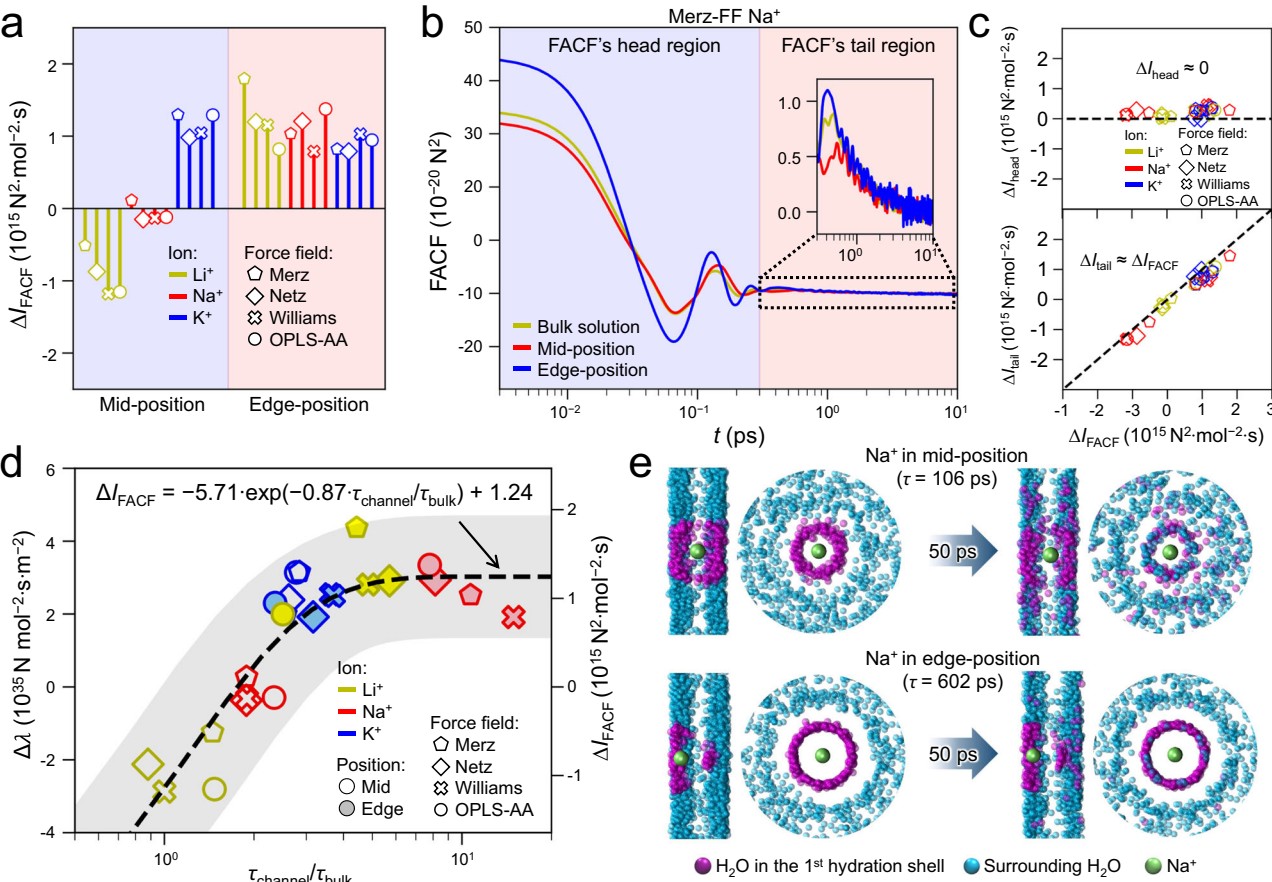

**Fig. 2 | Difference between ion-water friction in 2D nanochannels and bulk water: higher ratio $\tau_{channel}/\tau_{bulk}$ leads to larger $\Delta I_{FACF}$. a** $\Delta I_{FACF}$ ($I_{FACF}^{channel} - I_{FACF}^{bulk}$) of Li$^+$, Na$^+$ and K$^+$ at mid-position (blue area) or edge-position (red area). **b** FACFs of Na$^+$ (simulated with Merz FF) in different solvation environments, bulk solution (green), mid- (red) or edge- (blue) position. These FACFs all oscillate violently in the head part, then smoothly decay to 0 in the tail part. See Supplementary Figs. 15–18 for FACFs of other ions. **c** Compared with bulk solution, $I_{head}$ changes slightly ($\Delta I_{head} = I_{head}^{channel} - I_{head}^{bulk} \approx 0$), thus $I_{tail}$ dominates the change of $I_{FACF}$ ($\Delta I_{tail} = I_{tail}^{channel} - I_{tail}^{bulk} \approx \Delta I_{FACF}$). **d** $\Delta\lambda$ (or $\Delta I_{FACF}$) correlates with $\tau_{channel}/\tau_{bulk}$. $\Delta\lambda$ is the difference between ion-water friction coefficient ($\lambda$) in nanochannel and that in bulk

solution, which equals to $\Delta I_{FACF}$ multiplied by a constant, $\frac{1}{\gamma k_B T}$ (see Eq. (4) in "Methods"). The black curve is exponential fitting result. The dash curve represents the quantitative correlation between $\Delta I_{FACF}$ (unit: $10^{15}$ N$^2 \cdot$mol$^{-2} \cdot$ s) and $\tau_{channel}/\tau_{bulk}$: $\Delta I_{FACF} = -5.71 \cdot \exp(-0.87 \cdot \frac{\tau_{channel}}{\tau_{bulk}}) + 1.24$ and the shadowed area represents the prediction error. **e** Exchange of water molecules between the 1$^{st}$ HS of Na$^+$ (simulated with Merz FF) and the surrounding water layers. 100 snapshots of a 1 ns simulation trajectory are superimposed. In each sub-figure, the left one is the side view, while the right is top view. For clarity, only the upper water layer is shown in the top view. When $\tau$ is small, water molecules exchange rapidly, while they exchange slowly when $\tau$ is large. Source data are provided as a Source Data file.

component of ion-water interaction, driving water molecules in the middle of HS toward the channel-walls for energetic reasons (Supplementary Note 4 and Supplementary Figs. 19–20). For instance, the ion-water interaction for K$^+$ is so weak (insufficient to maintain a spherical HS) that its HS is distorted to two rings when locating at mid-position (left of Fig. 3a). In addition, HSs of ions at edge-position become too crowded to allow water molecules locate in regions between the ring and the pole (right of Fig. 3a, Supplementary Note 4). When comparing an ion at different solvation environments, the density of HS ($\rho_{HS}$) of the distorted HS usually increases, as its volume decreases yet the hydration number hardly changes (Supplementary Figs. 2–5). Thus, the free energy profile of water molecules around the ion was calculated from the density profile with Eq. (7) (Fig. 3b, Supplementary Fig. 21), and another correlation emerges: $\ln(\tau_{channel}/\tau_{bulk})$ decreases linearly with $\Delta F_{channel} - \Delta F_{bulk}$ (Fig. 3c), where $\Delta F$ is the free energy difference between water in HS and water in the surrounding solvent (e.g. the water layers in nanochannel, or bulk water in the free solution), and the subscript (channel or bulk) indicates ions located in nanochannel or in bulk solution, respectively. Such correlation indicates that higher free energy of water molecules in HS (relative to surrounding solvent) helps water molecule to escape from the HS, which decreases $\tau$.

## Mechanism of the ion motion in 2D nanochannels

The physical nature of the $D_{channel}/D_{bulk}$ - $d_{ion-wall}$ correlation and the ion-water friction can be explained as follows (Fig. 4a). When an ion approaches a water layer in 2D nanochannel (smaller $d_{ion-wal}$) or has a large $r_{HS}$ (like K$^+$), the nanoconfinement effect significantly distorts its HS (Supplementary Note 4). This leads to a larger $|\Delta F|$ between the HS and the water layer, which hinders water molecules to exchange between them, and $\tau$ significantly increases. Thus, the decay of FACF becomes slower with increasing $I_{FACF}$, i.e. the ion-water friction rises, and thus $D_{channel}$ decreases consequently.

Such mechanism well explains all above findings. For instance, when Na$^+$ (or other small ions) moves from mid- to edge-position, its sphere-like HS becomes distorted to a ring and a pole (Fig. 3a), with $\rho_{HS}$ increasing from 17.3 g·cm$^{-3}$ to 44.2 g·cm$^{-3}$ (see Supplementary Note 4 for explanation) and $|\Delta F|$ increasing by ~2.5 kJ·mol$^{-1}$ (Fig. 3b). This leads to an increasing ratio $\tau_{channel}/\tau_{bulk}$, and $\Delta I_{FACF}$ increases up to ~1000 × 10$^{12}$ N$^2$·mol$^{-2}$ · s (Fig. 2d), which results in a slower ion diffusion due to rised friction. Therefore, $D_{channel}/D_{bulk}$ decreases with decreasing $d_{ion-wall}$ (Fig. 1b). Contrarily, when K$^+$ moves from mid-position to edge-position, the $\rho_{HS}$ and $\Delta F$ change only very slightly (Fig. 3b), which gives a constant $D_{channel}/D_{bulk}$ (Fig. 1b). Such mechanism also explains why ions with large $r_{HS}$ usually show a negative

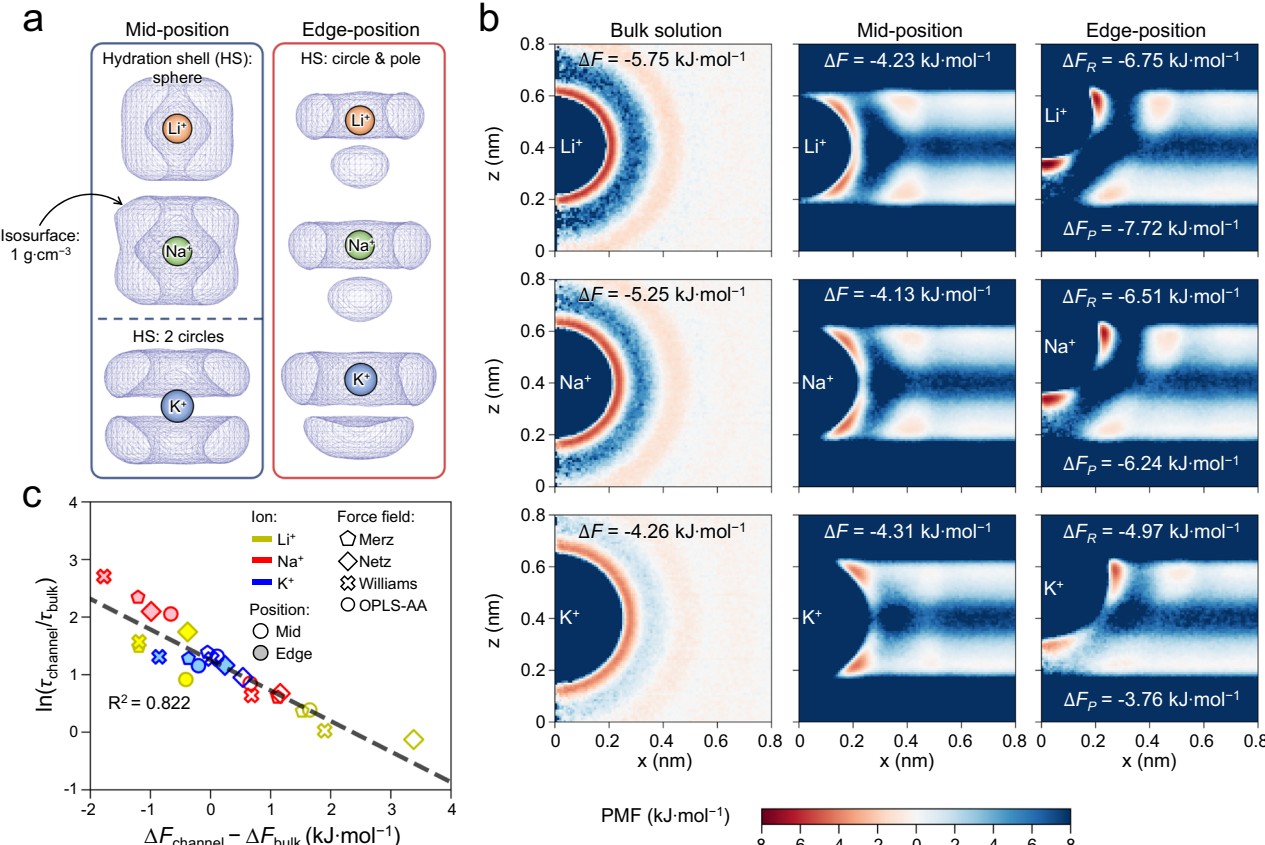

**Fig. 3 | High free energy in the HS helps water molecules to escape, resulting in a small $\tau_{channel}/\tau_{bulk}$. a** Spatial distribution functions (SDFs) for water molecules in the 1st hydration shells (HS) of Li⁺, Na⁺ and K⁺ at the mid- or edge-position in a 2D nanochannel. At the mid-position, Li⁺ and Na⁺ possess sphere-like HSs while the K⁺ HS splits into two rings. At the edge-position, the HSs of all three ions are distorted to rings and poles. **b** Potential of mean force (PMF) profiles of water molecules around Li⁺, Na⁺ and K⁺ in bulk solution, mid- or edge-position of the 2D nanochannels, simulated with Merz FF. $\Delta F$ refers to the free energy difference between the ion HS and the surrounding (e.g. the water layers in nanochannel, or bulk water

in the bulk solution), with subscripts 'R' and 'P' referring to the ring and pole part of HS, respectively. **c** $\ln(\tau_{channel}/\tau_{bulk})$ correlates linearly with $\Delta F_{channel}-\Delta F_{bulk}$: $\ln\left(\frac{\tau_{channel}}{\tau_{bulk}}\right) = \frac{-0.00443(\Delta F_{channel}-\Delta F_{bulk})}{RT} + 1.265$, where R is the ideal gas constant. Note at $\Delta F_{channel}-\Delta F_{bulk} \approx 0$, $\tau_{channel}/\tau_{bulk}$ is yet above 1, which can be attributed to the nanoconfinement effect of 2D nanochannels, i.e. the channel walls prevent water molecules from leaving the HS along the channel height direction, thus hindering water molecules to exchange between HS and surrounding solvent in nanochannel[52]. Source data are provided as a Source Data file.

deviation (Fig. 1b), particularly for location with large $d_{ion-wall}$. That is, large $r_{HS}$ leads to distorted HS, larger $|\Delta F|$, increased $\tau$ and $I_{FACF}$, and consequently, decreased $D_{channel}/D_{bulk}$ (Fig. 4a). Therefore, when $d_{ion-wall}$ is large, $D_{channel}/D_{bulk}$ of large-$r_{HS}$ ions (independent on $d_{ion-wall}$) may be quite smaller than the prediction of $D_{channel}/D_{bulk} \sim d_{ion-wall}$ correlation, whose slope is positive (Fig. 1b).

Besides the self-diffusion of hydrated ions, the ion migration behavior under external electric field in the confined 2D nanochannels are also investigated. We simulated the ion electromigration in 2D nanochannels constructed by graphene, h-BN, g-C₃N₄ or MoS₂ under electric field between 0.1 and 0.5 V·nm⁻¹ along the x direction (Fig. 4b, Supplementary Figs. 22, 23), and found that the ion mobility $\mu$ (independent on electric field, see Supplementary Fig. 24) is proportional to their self-diffusivity $D_{channel}$, which agrees well with the Nernst−Einstein relation: $D = k_BT\mu/q$, where $k_B$ is Boltzmann constant, $T$ is temperature and $q$ is ion's charge. Combining simulations and theoretical derivations (Supplementary Note 5), the validity of Nernst−Einstein relation in 2D nanochannel is proved for the first time. Such validity could be attributed to that, the electric field hardly changes the structure of water layers (Supplementary Fig. 25), in which the ions diffuse (without external field) or drift (driven by the electric field). Furthermore, the relationship between $\mu_{channel}/\mu_{bulk}$ and $d_{ion-wall}$ is similar to the ratio of the self-diffusivities without external field (Fig. 4c), indicating that the above mechanism (Fig. 4a) applies for both

ion self-diffusion (Fig. 1b) and electromigration (Fig. 4c) in 2D nanochannels. Please refer to Supplementary Table 8 for all studied systems and resulting conclusions in this work.

## Discussion

Our research provides a complete theoretical framework for the transport of hydrated monatomic ions in various 2D nanochannels assembled by graphene, h-BN, g-C₃N₄, MoS₂, including self-diffusion and electromigration. Through in-depth MD simulations employing multiple force fields, we unveil that the diffusivity of ions in a 2D nanochannel $D_{channel}$ depends on their HS radius ($r_{HS}$) and the position in the nanochannel, differing from their constant bulk diffusivity $D_{bulk}$. In detail, the ratio of ion diffusivity $D_{channel}/D_{bulk}$ shows for ions with small $r_{HS}$ a linear correlation with $d_{ion-wall}$, and for ions with large $d_{ion-wall}$, $D_{channel}/D_{bulk} > 1$ is found. In contrast, for ions with large $r_{HS}$ (e.g., K⁺, Rb⁺, Cs⁺) $D_{channel}/D_{bulk}$ is constant, independent on $d_{ion-wall}$. Importantly, the underlying physical mechanism has been revealed with quantitative correlations, which bridges fundamental physical quantities such as the free energy profile of water molecules around the ion, the residence time of water molecule in HS and the water-ion friction force. Moreover, besides the ions' self-diffusivity, their mobility under an electric field also follows the above rule, and the well-known Nernst−Einstein relation is proved to be valid for electrolytes in 2D nanochannels. To our knowledge, this work provides a universal and general approach to quantitatively study the ion

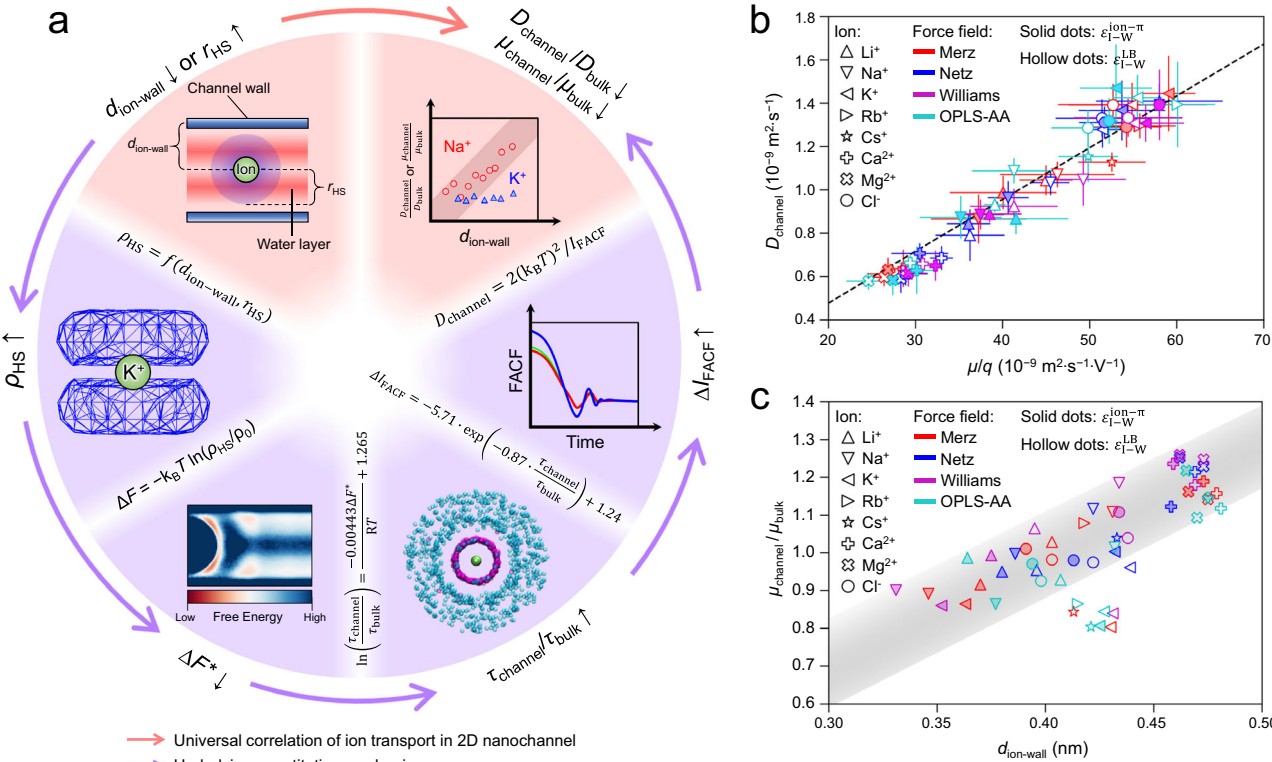

**Fig. 4 | Ion transport mechanism in 2D nanochannels and the involved quantitative relations. a** Underlying mechanism how water layers and HSs affect ion transport in 2D nanochannels. The quantitative correlations between involving physical quantities are listed where applicable, also discussed in previous discussions on Figs. 2, 3 and Eq. (8) in "Methods" section. $\Delta F^*$ refers to $\Delta F_{\text{channel}}$-$\Delta F_{\text{bulk}}$. As for some physical quantity such as $\tau$, $\Delta F$, the difference between this quantity in the nanochannel and the bulk water ($\tau_{\text{channel}}/\tau_{\text{bulk}}$ and $\Delta F_{\text{channel}}$-$\Delta F_{\text{bulk}}$) is more frequently discussed than the quantity itself, so as to focus on how water layers in 2D nanochannels affect these quantities. **b** Linear relation between ion diffusivity $D_{\text{channel}}$ and mobility $\mu$ under electric field E = 0.1 V·nm$^{-1}$ in graphene 2D

nanochannel (see Supplementary Figs. 22–23 for h-BN, g-C$_3$N$_4$ and MoS$_2$ nanochannels). The black line is the linear fitting result (y = 0.0239·x, R$^2$ = 0.9683), whose slope is consistent with the theoretical value of 0.0257 J · C$^{-1}$ in the Nernst–Einstein relation. Error bars represent the standard error ($n$ = 3 independent MD simulations), and the centers of error bars indicate the means. **c** Under E = 0.1 V·nm$^{-1}$, $\mu_{\text{channel}}/\mu_{\text{bulk}}$ increases linearly with $d_{\text{ion-wall}}$ for all ions and FFs under study (shadowed area represents the prediction error), except ions with large $r_{\text{HS}}$ (e.g. K$^+$, Rb$^+$, Cs$^+$), which is similar to the $D_{\text{channel}}/D_{\text{bulk}}$ ~ $d_{\text{ion-wall}}$ correlation in Fig. 1b. Source data are provided as a Source Data file.

transport in confined 2D nanochannels tracing back to the physical nature. The position-dependent diffusivity/mobility of ions confined in the studied 2D nanochannels is to the sharp contrary of bulk solution, where ions' diffusivity/mobility are constant. This indicates ions' transportation in 2D nanochannels could be adjusted by controlling their position by some external fields[4], e.g. electric- or magnetic-field, which may be exploited for various nano-devices[5–7,19]. Furthermore, considering the variation range of diffusivity (or mobility) shown in Figs. 1 and 4, more attention should be put on the entrance effect[17] or specific ion-wall attraction[50] when employing 2D nanochannels for ion-sieving. In brief, all the findings in our work pave the way to thoroughly understand the ion transport phenomena in nanochannels, also help design and construction of high-performance 2D nanochannels for a wide range of application fields, such as ion-sieving, nano-device, osmotic energy conversion, etc.

## Methods
### MD simulations
MD simulations were performed for the 2D nanochannel systems shown in Fig. 1a: salt solution containing one ion (Li$^+$, Na$^+$, K$^+$, Rb$^+$, Cs$^+$, Ca$^{2+}$, Mg$^{2+}$ or Cl$^-$) and a certain number of water molecules (see later explanation and Supplementary Table 8) is confined in a 2D nanochannel whose wall material is graphene, h-BN, MoS$_2$ or g-C$_3$N$_4$ (Supplementary Fig. 6). The x-y size of the simulation box equals to that of different wall materials (see Supplementary Table 8). In an 2D nanochannel, the distance between the channel walls is 1 nm

(Supplementary Fig. 6). Note in MoS$_2$ nanochannel, it refers to the distance between the 2 inner S atom planes. The atoms of channel walls were fixed, which have yielded accurate simulation results as plenty of recent publications[51,52] did.

The number of water molecules ($N_{\text{water}}$) in 2D nanochannels shown in Fig. 1a was determined by simulating water-soaked 2D nanochannel system[51]: a 2D nanochannel (identical with that in Fig. 1a) placed in the center of a simulation box containing 15,723 water molecules (box size: about 10 nm × 10 nm × 5 nm). $N_{\text{water}}$ was calculated as $N_{\text{water}} = \langle N_c \rangle \cdot A/A_c$, where $A$ is the x-y area of 2D nanochannel (e.g., 4.920 × 5.124 nm$^2$ for graphene 2D nanochannel; See Supplementary Table 8); $A_c = 3 × 3$ nm$^2$ is the x-y area of center region of the 2D nanochannel in water-soaked 2D nanochannel system; $\langle N_c \rangle$ is the ensemble-averaged number of water molecules in the center region of water-soaked 2D nanochannel (whose x-y area is $A_c$).

MD simulations for an ion in bulk solution were also performed using a cubic simulation box (size: 2.55 nm × 2.55 nm × 2.55 nm). The simulation system consists of 1 ion and 553 water molecules (corresponding to the bulk water density of 0.998 g/cm$^3$, excluding the mass of ion). The simulation systems in our work usually contain 1 ion, if not otherwise specified, to avoid the interference of ion-ion correlations, so that we could focus on the ion-water interactions.

The intermolecular interactions include coulombic and van der Waals interactions, where van der Waals interactions were described as the Lennard-Jones (LJ) potential. If not otherwise specified, LJ parameters were derived from Lorentz-Berthelot (LB) mixing rule. SPC/E water

model[53] was employed in this work; the force field (FF) parameters for graphene carbon atoms ($\sigma_C = 0.3214$ nm, $\varepsilon_C = 0.4900$ kJ·mol$^{-1}$) were taken from the literature[54]. Four sets FF parameters for ions from Netz et al.[42,43], Merz et al.[40,41], Williams et al.[44] (referred to as Netz FF, Merz FF, and Williams FF, respectively) and OPLS-AA[39] FF were employed in this work (Supplementary Table 1), which well reproduced the experimental diffusivity of ions in bulk solutions (Supplementary Fig. 26). Our simulations also reproduced the experimental ion mobility (Supplementary Fig. 27), both in bulk solutions and 2D nanochannels[15]. For the LJ parameters between ions and graphene, two versions of FFs (Supplementary Table 2) were employed: (i) original FFs: calculated with the LB mixing rule (denoted as $\varepsilon_{I-W}^{LB}$); (ii) optimized FFs: the recently optimized LJ parameters[44,45] (denoted as $\varepsilon_{I-W}^{ion-\pi}$) which describe the ion-π interaction between ion and graphene accurately (LJ parameters, $\sigma_{I-W}$, were all calculated with the LB mixing rule). The LJ parameters and atomic charges for wall atoms of h-BN, MoS$_2$ and g-C$_3$N$_4$ were taken from refs. 55–57. (Supplementary Table 1).

The simulation trajectories were integrated via the leapfrog algorithm with a time step of 2 fs with periodic boundary conditions (PBCs) applied to x and y directions (the channel wall plane). For simulations of water-soaked 2D nanochannel system and bulk solution system, PBCs were applied to x, y and z directions. If not otherwise specified, the simulation setups for simulations of water-soaked 2D nanochannel system and bulk solution system were identical to the simulations of 2D nanochannels shown in Fig. 1a. The intermolecular LJ interactions were computed with a cutoff of 1.2 nm with tail correction. The long-range electrostatic interactions were calculated by the particle mesh Ewald (PME) method[58], with a real space cutoff of 1.2 nm. The temperature was controlled at 298.15 K by Nose-Hoover thermostat[59,60]. The length of bonds involving H atoms was constrained by SETTLE algorithm[61]. In a typical MD simulation procedure, an energy minimization for 10,000 steps with a steepest-descent minimization algorithm was performed, followed by a 2 ns NVT MD simulation increasing the temperature to 298.15 K; subsequently, a 55 ns production simulation was performed in NVT ensemble and the last 50 ns data were used for analysis. For water-soaked 2D nanochannel system, a 10 ns production simulation was performed in NPT ensemble (the pressure was controlled at 1 bar by Parrinello-Rahman barostat) and the last 5 ns data were used for calculating $N_{water}$. All MD simulations were performed by GROMACS 5.1.5 simulation package[62] and VMD software[63] was utilized for visualization.

## Simulation data analyses

**Average ion-wall distance ($d_{ion-wall}$).** We have calculated $d_{ion-wall}$ as follow:

$$d_{ion-wall} = \int_{-0.5}^{0.5} \rho_i(z)(0.5 - |z|)dz \tag{1}$$

where $z$ values of +/- 0.5 and 0 corresponds to the position of the bottom/upper channel wall (the inner S planes for MoS$_2$ channel wall), the middle of the nanochannel, respectively, $\rho_i(z)$ is ion's distribution probability along z direction (Supplementary Figs. 1, 7), 0.5-|z| is ion's distance from the nearest channel wall. The average (water's) oxygen-wall distance was calculated similarly, show in Fig. 1b.

## Diffusivity calculated with the mean square displacement (MSD) method

Ion's diffusivity ($D$) could be obtained by computing the slope of its MSD curve[51]:

$$D = \lim_{t\to\infty} \frac{\left\langle |\mathbf{r}(t) - \mathbf{r}(0)|^2 \right\rangle}{\beta \cdot t} \tag{2}$$

where $\mathbf{r}(t)$ stands for ion's position at time $t$, $\left\langle |\mathbf{r}(t) - \mathbf{r}(0)|^2 \right\rangle$ is the ensemble average of ion's MSD for a given simulation time $t$, the coefficient $\beta$ is 4 for ion in 2D nanochannel or 6 for ion in bulk water. For $D$ calculation, we performed 5 independent 55-ns production simulations, with averaged results and stand error reported. Diffusivities in this work were usually calculated with the MSD method if not otherwise specified.

## Diffusivity calculated with the force auto-correlation function (FACF) method

We also obtained $D$ using the FACF method[64]:

$$FACF(t) = \left\langle F(t)F(0) \right\rangle \tag{3}$$

$$D = \frac{k_B T}{\lambda} = \frac{\gamma k_B^2 T^2}{\int_0^\infty FACF(t)dt} \tag{4}$$

where $\left\langle F(t)F(0) \right\rangle$ is the Force Auto-Correlation Function (FACF); $F(t)$ is the total force that water molecules exert on the ion at time $t$, respectively; <...> represents the ensemble average; $\lambda = \frac{1}{\gamma k_B T}\int_0^\infty FACF(t)dt$ is the ion-water friction coefficient (the coefficient $\gamma$ is 2 for ion in 2D nanochannel or 3 for ion in bulk solution); $k_B$ and $T$ are Boltzmann constant and temperature, respectively. In FACF simulations, the ion was fixed so that the 'infinite mass' approximation could apply[65]. Note as for simulations elsewhere in this work, the ion was free to move.

## Residence time of water molecules in ions' 1st hydration shell

The residence time ($\tau$) for water molecules staying in ion's 1st HS was calculated by the residence auto-correlation function $R(t)$[66]:

$$\tau = \int_0^\infty R(t)dt \tag{5}$$

$$R(t) = \frac{1}{N_h}\sum_{i=1}^{N_h} \left\langle \theta_i(t)\theta_i(0) \right\rangle \tag{6}$$

where $\theta_i(t)$ is the Heaviside function for the ith water molecule in ion's 1st HS, which is 1 if the water molecule's O$_w$ atom stays in the 1st HS at time $t$ and 0 otherwise.

## Potential of mean force (PMF)

The PMF profiles for water molecules around the ion were calculated as follow[67,68]:

$$PMF(r, z) = -k_B T \ln \frac{\rho_{(r,z)}}{\rho_0} \tag{7}$$

where $r$ and $z$ are the radial distance and z-direction distance from the ion, respectively; $k_B$ and $T$ are Boltzmann constant and temperature, respectively; $\rho_{(r,z)}$ is the local water density in $(r, z)$ position; $\rho_0$ represents bulk water density (1 g·cm$^{-3}$) for the ion in bulk water or the peak density of water layers (3.27 g·cm$^{-3}$) for the ion in nanochannel. The free energy difference ($\Delta F$) between HS of an ion and the surrounding solvent was calculated as:

$$\Delta F = -k_B T \ln \frac{\rho_{HS}}{\rho_0} \tag{8}$$

where $\rho_{HS}$ is the peak water density (maximum $\rho_{(r,z)}$) in the HS of the ion.

## Ion mobility

Ion's mobility ($\mu$) can be calculated as:

$$\mu = \frac{v}{E} = \frac{1}{E} \cdot \lim_{t \to \infty} \frac{\langle |\mathbf{r}(t) - \mathbf{r}(0)| \rangle}{t} \tag{9}$$

where $v$ is ion's speed, $E$ is the strength of electric field (along the x direction for ions in 2D nanochannel); $\mathbf{r}(t)$ stands for ion's position at time $t$ and $\langle |\mathbf{r}(t) - \mathbf{r}(0)| \rangle$ is the ensemble average of ion's displacement for a given simulation time $t$.

## Data availability

Source data are provided with this paper. The source data produced in this study have been deposited in Figshare [https://figshare.com/s/0352095f89ab307bd783]. Source data are provided with this paper.

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

## Acknowledgements

Y.W. acknowledges the funding from the National Natural Science Foundation of China (U23A20115). L.L. acknowledges the funding from Science and Technology Key Project of Guangdong Province (2025B0101060003), Guangdong Natural Science Foundation (2024A1515012725) and the National Natural Science Foundation of China (22078104). Y.W. also acknowledges the funding from Guangdong Natural Science Foundation (2024A1515012724), Guangzhou Science and Technology Project (2024A04J6251), State Key Laboratory of Pulp and Paper Engineering (2024ZD03).

## Author contributions

S.L., Y.L. and L.L. performed molecular dynamics simulations and analyzed data. L.L., Y.W. and H.W. jointly supervised this work. S.L., L.L., L.D., Y.W. and H.W. contributed to discussions and manuscript preparation.

## Competing interests

The authors declare no competing interests.
