## [Transparent Peer Review file · Nature Communications]

Theoretical Framework for Confined Ion Transport in Two-dimensional Nanochannels

Corresponding Author: Professor Haihui Wang

Version 0:

Reviewer comments:

Reviewer #1

(Remarks to the Author)

In this work, the authors proposed a theoretical framework for self-diffusion and electromigration of hydrated monatomic ions in various nanochannels. It thoroughly reveals the mechanism of ionic transport in confined 2D nanochannels, tracing back to the most fundamental physical nature. Such mechanism has seldom been reported, yet it was established on solid data and clear theoretical derivation by the authors. It clearly explains various recent experimental phenomena, e.g. ref 17, also thoroughly validates other very important physical rules in nanochannels, which has been taken for grant. The distinct difference between the mechanism in nanochannels and that in bulk solution is fully revealed. This work is a considerable advance on understanding the ions' motion in confined nanochannels, a long standing yet very important scientific question. This work surely will promote the developments in a large variety of scientific or application fields. I think this is a very interesting and meaningful work, which is well organized and written. I recommend it to be published on Nature Communications after addressing the following comments.

1. This work shows huge data from several studied nanochannel systems. These systems contain various channel wall materials, various external fields or restrains, which confirm that, the revealed mechanism is quite universal. Could the authors list the information of all their studied systems in a table? e.g. the channel wall materials, the water molecule numbers, any external fields or restrains... It would be even better that, if the authors can also indicate what principle/mechanism did they obtain from which systems in the table. I believe such table could surely help the readers understand such huge data in the work.
2. The authors demonstrated that "For the first time, the mechanism of ionic transport through confined 2D nanochannels, distinct from bulk solution, has been traced back thoroughly to its physical nature". Personally, I think this assertion is too arbitrary and ignores the contribution from recent literature. For example, Journal of Physical Chemistry C, 123: 1462–1469, 2019; Nanoscale, 11: 8449–8457, 2019.
3. There are numerous data in Fig. 1b, which were produced with various force field. What are the optimized force fields (FFs) in Fig. 1b, can the authors show the refs in the caption? In addition, can the authors use solid or hollow dots to distinguish 'original' and 'optimized' FFs, instead of use different colors for now? What is the ratio, $D_{\text{water-in-channel}}/D_{\text{water-bulk}}$ (D means diffusivity)? Can the authors show this data in Fig. 1b for comparison?
4. The authors studied several FFs, which all re-produced the experimental diffusivity of ions in bulk solution. I'm wondering how these FFs affect the ions' diffusivity in the studied nanochannels? Can the authors study D-ion-channel instead of the ratio $D_{\text{ion-channel}}/D_{\text{ion-bulk}}$ to see what results they could obtain? This could even help the readers working on ions' motion in bulk solution or nanochannels.
5. The authors thoroughly validated Nernst-Einstein relation in 2D nanochannels with both simulation data and theoretical derivations, which would considerably help further research in relevant field. I'm wondering if the authors can validate Nernst-Einstein relation for varying the Electric field? i.e. take a certain ion, and vary the E-field, is the mobility a constant?
6. The authors simulated one ion in the nanochannel to avoid the ion-ion correlation, and obtain much more accurate results for ion-water interaction, the main point of this work. The authors are suggested to add such point in 'Methods'.
7. Can the authors add more significant figures to 1 g/cm^3 in 'Methods'?
8. When they calculated bulk water density, did they exclude the ion?

Reviewer #2

(Remarks to the Author)

First of all, it is nearly impossible from what is written in this manuscript to tell what model is being simulated. For example, I presume that the labels Merz FF, etc. refer to a model studied by Merz, etc., whose parameters are given in table I in the supplementary material, and the results of simulations with these models are given in table II of the supplementary materials? What does FF stand for? The answers to these questions should be given in the text of the article. Also, I did not see any place in either the article or the supplementary material where the various Lennard-Jones parameters, i.e., the headings of table I in the supplementary material ,etc. , are defined, or are these parameters defined in the references to the models referred to a Merz FF, etc.? In that case, the authors should say so in the paper. Otherwise, it is impossible to determine what is actually being simulated.

Fig. 1c seems to say that depending on whether the ion interacts with the carbon atom with a Lennard-Jones interaction with one of the two energy parameters, the distribution can be peaked on the center of the channel or displaced to sides of the channel, or am I misunderstanding what is being shown in this plot?

Plotting the residence time of hydration shell water as a function of the force-force correlation function is somewhat confusing since the physical quantity is the diffusion constant, which is proportional to the inverse of the force-force correlation function, or the friction coefficient, which is proportional to the force-force correlation function. So, perhaps it would be better to plot the residence time as a function of the diffusion constant or the friction coefficient.

In Li, et. al., Nat. Nanotechnol. 18, 177 (2022), which studies potassium ion flow through carbon nanotubes, the attractive interaction between the ion and the nanotube makes up for the increase of the hydrated ion's energy resulting from the removal of water molecules, which allows the ion to enter the nanotube. For the flat surfaces in these simulations, there does not seem to be any loss of hydration shell water molecules. I presume this is because for flat spaces, there is space in the directions parallel to the surfaces to accommodate the water molecules that must move out of the way to allow the hydrated ion to enter the channels? Does the friction increase because there are fewer water molecules between the ion and the surfaces?

Are there experimental results to compare these results to?

Reviewer #3

(Remarks to the Author)

This is a very interesting simulation study where the self diffusion and electrophoretic mobility of ions in nanochannels has been thoroughly investigated. The main strength of this paper is that different ions corresponding to the extremes of the Hofmeister series have been compared and that mono- as well as divalent ions have been studied. Also, different ionic force fields have been compared and the results are robust with respect to a change of the force field. As a main result, a correlation between ion position and ion diffusivity has been found.

1) What is called Stokes-Einstein relation on page 4 is in fact the Einstein relation.

2) The friction coefficient is extracted using the integral over the force-autocorrelation function, which only gives a converging result in the limit of an immobilized ion (or an ion with infinite mass), as correctly pointed out by the authors in the Supplement. However, the friction coefficient of an object is modified by confinement, as was recently shown ("External Potential Modifies Friction of Molecular Solutes in Water" Daldrop et al, PHYSICAL REVIEW X 7, 041065 (2017)). This effect is due to the effect that the water exchange dynamics around an object depends on whether the object is free to move in response to the hydration water dynamics or not. In fact, the ion friction coefficient has been shown to increase due to confinement by about 5% for a negative ion and by about 20% for a positive ion ("Memory-kernel extraction for different molecular solutes in solvents of varying viscosity in confinement" Kowalik et al, PHYSICAL REVIEW E 100, 012126 (2019)). The authors should have observed these modifications in their simulations, a discussion of this is in order. The alternative would be to obtain the friction from the memory kernel, which however is rather involved to extract. I do not believe that the mechanism for the friction changes due to this confinement effect, but this should be pointed out clearly in the paper.

Version 1:

Reviewer comments:

Reviewer #1

(Remarks to the Author)

In this revised version, the authors have addressed all my concerns and the manuscript has been improved significantly. Now I can recommend its publication.

Reviewer #2

(Remarks to the Author)

The authors appear to have answered all of my questions. As far as I am concerned, you should publish this manuscript

Reviewer #3

(Remarks to the Author)

I carefully read the reply letter and the revised paper and conclude that the authors have adequately addressed my comments and also the comments by the other referees. I therefore recommend publication of the paper as is.

Prof. Haihui Wang
Changjiang Chair Professor
NSFC outstanding Young Investigator
Fellow of The Royal Society of Chemistry
Department of Chemical Engineering
Tsinghua University
Beijing 100084, China
Tel.: +86-10-62793144
E-mail: cehhwang@tsinghua.edu.cn

Response to the Reviewers' Comments

Many thanks to the reviewers for their valuable comments and suggestions. The followings are the point-by-point answers to the concerns:

Response to Reviewer 1

In this work, the authors proposed a theoretical framework for self-diffusion and electromigration of hydrated monatomic ions in various nanochannels. It thoroughly reveals the mechanism of ionic transport in confined 2D nanochannels, tracing back to the most fundamental physical nature. Such mechanism has seldom been reported, yet it was established on solid data and clear theoretical derivation by the authors. It clearly explains various recent experimental phenomena, e.g. ref 17, also thoroughly validates other very important physical rules in nanochannels, which has been taken for grant. The distinct difference between the mechanism in nanochannels and that in bulk solution is fully revealed. This work is a considerable advance on understanding the ions' motion in confined nanochannels, a long standing yet very important scientific question. This work surely will promote the developments in a large variety of scientific or application fields. I think this is a very interesting and meaningful work, which is well organized and written. I recommend it to be published on Nature Communications after addressing the following comments.

Response: Thanks for your encouragement and kind comments. We have revised our manuscript carefully according to your suggestions point-by-point, and we hope these added simulations and explanations will help readers understand our work more easily.

1. This work shows huge data from several studied nanochannel systems. These systems contain various channel wall materials, various external fields or restrains, which confirm that, the revealed mechanism is quite universal. Could the authors list the information of all their studied

systems in a table? e.g. the channel wall materials, the water molecule numbers, any external fields or restrains... It would be even better that, if the authors can also indicate what principle/mechanism did they obtain from which systems in the table. I believe such table could surely help the readers understand such huge data in the work.

Response: Thanks for your kind consideration and valuable suggestions. We have added a new table (Table R1) to better present the extensive data in our work and to highlight the revealed mechanisms. Table R1 lists all studied systems, including channel wall materials, the numbers of water molecule and ion, external fields, constraints, as well as the specific mechanisms or principles derived from them.

Action: According to your comments, Table R1 has been added in the revised Supplementary Information (SI) as Supplementary Table 8 on page 43, and the main text has also been revised to refer to this Table on page 8, highlighted in yellow.

Table R1 | Simulation systems and resulting principles for ion transport in 2D nanochannels or in bulk solution.

Channel wall material	Ion	Ion concentration	Electric field (V/nm)	Constraint on ion	Principles
Graphene	Li ⁺ , Na ⁺ , K ⁺ , Rb ⁺ , Cs ⁺ , Ca ²⁺ , Mg ²⁺ , Cl ⁻	Infinite dilution: 1 ion + 553 water molecules ^[a]	0	No constraint	$D_{channel}/D_{bulk} \sim d_{ion-wall}$ correlation (Fig. 1b-1d, red parts in Fig. 4a)
				harmonic potential	$D_{channel}/D_{bulk} \sim d_{ion-wall}$ correlation (Supplementary Fig. 8)
				Ion location fixed ^[c]	Quantitative mechanisms of “ $\rho_{HS} \rightarrow \Delta F \rightarrow \tau \rightarrow I_{facf}$ ” (Figs. 2&3, blue parts in Fig. 4a) FACF method yields $D_{channel}/D_{bulk}$ (or D) values consistent with those yielded by MSD method. (Supplementary Figs. 12&13)
		0.1 M: 4 Cations + 4 Cl ⁻ + 2212 water molecules ^[a]	0.1 ~ 0.5	No constraint	Nernst–Einstein relation holds true for ions in 2D nanochannels (Fig. 4b, Supplementary Figs. 21) $\mu_{channel}/\mu_{bulk} \sim d_{ion-wall}$ correlation (Fig. 4c, red parts in Fig. 4a) Ion mobility keeps constant as electrical field increases from 0.1 to 0.5 V/nm. (Supplementary Fig. 23)
				No constraint	MD simulations reproduce the experimental ion mobilities of studied ions in graphene 2D nanochannel. (Supplementary Fig. 26)
				No constraint	MD simulations reproduce the experimental ion mobilities of studied ions in graphene 2D nanochannel. (Supplementary Fig. 26)
hBN MoS ₂ g-C ₃ N ₄	Li ⁺ , Na ⁺ , K ⁺ , Rb ⁺ , Cs ⁺ , Ca ²⁺ , Mg ²⁺ , Cl ⁻	Infinite dilution: 1 ion + 587 (hBN), 703 (MoS ₂) or 777 (g-C ₃ N ₄) water molecules ^[a]	0	No constraint	$D_{channel}/D_{bulk} \sim d_{ion-wall}$ correlation (Fig. 1e)
			0.1	No constraint	Nernst–Einstein relation holds true for ions in 2D nanochannels (Fig. 4b, Supplementary Figs. 22)
bulk solution	Li ⁺ , Na ⁺ , K ⁺ , Rb ⁺ , Cs ⁺ , Ca ²⁺ , Mg ²⁺ , Cl ⁻	Infinite dilution: 1 ion + 553 water molecules ^[b]	0	No constraint	MD simulations reproduce the experimental diffusivities of studied ions in bulk solution (Supplementary Fig. 25)
				Ion location fixed ^[c]	FACF method yields D_{bulk} values consistent with those yielded by MSD method. (Supplementary Figs. 12&13)
		0.1 M: 4 Cations + 4 Cl ⁻ + 2212 water molecules ^[b]	0.1	No constraint	MD simulations reproduce the experimental mobilities of studied ions in bulk solution. (Supplementary Fig. 26)

[a]: The number of water molecules in nanochannels were determined based on the equilibrium water density in a water-soaked nanochannel under the pressure of 1 bar, as shown in our previous works¹. The x-y size of these 2D nanochannels are 4.920×5.124 nm² (graphene), 5.260×5.206 nm² (hBN), 5.525×5.742 nm² (MoS₂) or 5.705×6.176 nm² (g-C₃N₄). In the simulations of 0.1 M ion concentration, we doubled the x-y size of the

graphene nanochannels to accommodate 4 cations and 4 Cl⁻ ions.

[b]: The number of water molecules corresponded to the bulk water density of 0.998 g/cm³. In the simulations of 0.1 M ion concentration, the size of simulation system was expanded to accommodate 4 cations and 4 Cl⁻ ions (the water density keeps as the same).

[c]: Ions in nanochannel were fixed either at middle or edge position (see main text for explanations) to calculate ρ_{HS} , ΔF , τ and I_{FACF} (see Supplementary Note 1 for calculation details)

2. *The authors demonstrated that “For the first time, the mechanism of ionic transport through confined 2D nanochannels, distinct from bulk solution, has been traced back thoroughly to its physical nature”. Personally, I think this assertion is too arbitrary and ignores the contribution from recent literature. For example, Journal of Physical Chemistry C, 123: 1462–1469, 2019; Nanoscale, 11: 8449–8457, 2019.*

Response & Action: Thanks for your helpful suggestion. We have carefully learnt these two literatures and cited them in the revised manuscript (page 1) as refs. 32&33, also we have revised the above sentence to “*In this work, the mechanism of ionic transport through confined 2D nanochannels, distinct from bulk solution, has been traced back thoroughly to its physical nature*” in the revised manuscript (page 1).

3. *There are numerous data in Fig. 1b, which were produced with various force field. What are the optimized force fields (FFs) in Fig. 1b, can the authors show the refs in the caption? In addition, can the authors use solid or hollow dots to distinguish ‘original’ and ‘optimized’ FFs, instead of use different colors for now? What is the ratio, $D_{\text{water-in-channel}}/D_{\text{water-bulk}}$ (D means diffusivity)? Can the authors show this data in Fig. 1b for comparison?*

Response & Action: The ‘optimized’ force fields stand for the optimized Lennard-Jones (LJ) interaction parameters between ion and graphene, as shown in Supplementary Table 2. We have revised the caption of Fig. 1 and added the refs in the caption accordingly. According to your suggestions, the data points for ‘original’ and ‘optimized’ FFs in Fig. 1b have been revised to hollow and solid dots, respectively (as shown in Fig. R1 below). The data point of water molecules, with $D_{\text{channel}}/D_{\text{bulk}} = 0.76$, has been added into Fig. 1b. Moreover, for ease of understanding, we have revised the labels of Fig. 1b, replacing ‘original’ and ‘optimized’ by ‘ ϵ_{I-W}^{LB} ’ and ‘ $\epsilon_{I-W}^{\text{ion}-\pi}$ ’, respectively, where ϵ_{I-W} is the LJ parameter between ion and wall atoms, subscript *LB* indicates the parameters are calculated from the Lorentz-Berthelot (LB) mixing rule, and ion- π indicates they are optimized to describe the ion- π interactions in solution accurately (these explanations have been added to the page 2 of main text).

Fig. R1 | $D_{channel}/D_{bulk}$ increases linearly with $d_{ion-wall}$ for all ions and FFs under study (shaded area represents the prediction error), except ions with large r_{HS} (e.g. K⁺, Rb⁺, Cs⁺). Merz^{2,3}, Netz^{4,5}, Williams⁶ FFs (named by the authors who developed them) and OPLS-AA⁷ FF were employed. The data for water molecules (black hollow diamond) is shown for comparison.

4. The authors studied several FFs, which all re-produced the experimental diffusivity of ions in bulk solution. I'm wondering how these FFs affect the ions' diffusivity in the studied nanochannels? Can the authors study $D_{ion-channel}$ instead of the ratio $D_{ion-channel}/D_{ion-bulk}$ to see what results they could obtain? This could even help the readers working on ions' motion in bulk solution or nanochannels.

Response: Thanks for your helpful suggestion. We have looked at $D_{channel}$ instead of the ratio $D_{channel}/D_{bulk}$ (Fig. R2) closely. It reveals that 1) $D_{channel}$ of Mg²⁺ and Ca²⁺ are usually larger than corresponding D_{bulk} , while $D_{channel}$ of other ions are usually smaller; 2) When comparing the original version of a FF (ϵ_{I-W}^{LB}) with the optimized one ($\epsilon_{I-W}^{ion-\pi}$), $D_{channel}$ of Na⁺ show the most significant change with corresponding distribution profiles also changing greatly; 3) D_{bulk} affects $D_{channel}$ to some extent, i.e. larger D_{bulk} usually leads to larger $D_{channel}$. Although $D_{channel}$ themselves reveal less information than $D_{channel}/D_{bulk}$, they yield some useful clues as follows. The above point 2 inspires us consider $d_{ion-wall}$, which describes the change of ion distribution profiles. Point 3 inspires us study the ratio $D_{channel}/D_{bulk}$, so that we could focus on the difference between nanochannel and bulk solution and how such difference affect the motion mechanism for various ions (the main point of our work), instead of discuss specific details of individual

FFs or ions. In this way, all the studied ions and FFs in our work could complete the puzzle, ‘ $D_{channel}/D_{bulk}$ of small ions ($r_{HS} < K^+$'s r_{HS}) increases with $d_{ion-wall}$, while that of bulky ions ($r_{HS} \geq K^+$'s r_{HS}) is independent on $d_{ion-wall}$ ’ (Fig. 1b).

Action: We have added Fig. R2 into revised SI as Supplementary Fig. 10 (page 18), also the above discussions into revised manuscript (page 4) and SI as the new Supplementary Note 2 (page 17).

Fig. R2 | $D_{channel}$ in graphene 2D nanochannel versus D_{bulk} for the studied ions simulated with different force fields. For each force field, two different sets of LJ parameters between ions and graphene, ϵ_{I-W}^{LB} and $\epsilon_{I-W}^{ion-\pi}$ (Supplementary Table 2) were employed. The dot line is the guide line of $y = x$.

5. The authors thoroughly validated Nernst-Einstein relation in 2D nanochannels with both simulation data and theoretical derivations, which would considerably help further research in relevant field. I’m wondering if the authors can validate Nernst-Einstein relation for varying the Electric field? i.e. take a certain ion, and vary the E-field, is the mobility a constant?

Response: Thanks a lot for your helpful suggestion. We have taken Na^+ with original or optimized Merz FF, and calculate the mobility for Electric field ranging from 0.1 to 0.5 V/nm, which are constant (Fig. R3). We have also simulated ions in graphene 2D nanochannels with Electric field of 0.3 V/nm, yielding mobility close to those of 0.1 V/nm. These further validate the Nernst-Einstein relation for different electric fields.

Action: We have added Fig. R3 to SI as Supplementary Fig. 23 (page 41), and revised the manuscript (page 8, paragraph 3 of the ‘Mechanism of the ion motion in 2D nanochannels’

section) accordingly.

Fig. R3 | The ion mobilities are constant for different electric fields. (a) In graphene 2D nanochannel, mobilities of all studied ions at $E = 0.3 \text{ V/nm}$ are consistent with those at 0.1 V/nm . (b) Mobility of Merz-FF Na^+ keeps constant under $E = 0.1 \sim 0.5 \text{ V/nm}$. Note the mobilities for Na^+ with $\epsilon_{I-W}^{ion-\pi}$ are lower than those with ϵ_{I-W}^{LB} in the nanochannel, as $d_{ion-wall}$ for Na^+ with $\epsilon_{I-W}^{ion-\pi}$ decreases (Figs. 1-3 in the main text and relevant discussions).

6. The authors simulated one ion in the nanochannel to avoid the ion-ion correlation, and obtain much more accurate results for ion-water interaction, the main point of this work. The authors are suggested to add such point in ‘Methods’.

Response & Action: According to your suggestion, we have added such point to the 2nd paragraph of ‘Methods’ section (page 10) as “The simulation systems in our work usually contain 1 ion, if not otherwise specified, to avoid the interference of ion-ion correlations, so that we could focus on the ion-water interactions”.

7. Can the authors add more significant figures to 1 g/cm^3 in ‘Methods’?

Response & Action: According to your suggestion, we have replaced ‘ 1 g/cm^3 ’ by ‘ 0.998 g/cm^3 ’ for water density with more significant figures in the revised manuscript (page 10) and highlighted in yellow.

8. When they calculated bulk water density, did they exclude the ion?

Response & Action: Yes, we excluded the ion when calculating bulk water density. According to your comment, we have added such information to the 2nd paragraph of ‘Methods’ in the revised manuscript (page 10) and highlighted in yellow.

Response to Reviewer 2

1. First of all, it is nearly impossible from what is written in this manuscript to tell what model is being simulated. For example, I presume that the labels Merz FF, etc. refer to a model studied by Merz, etc., whose parameters are given in table I in the supplementary material, and the results of simulations with these models are given in table II of the supplementary materials? What does FF stand for? The answers to these questions should be given in the text of the article. Also, I did not see any place in either the article or the supplementary material where the various Lennard-Jones parameters, i.e., the headings of table I in the supplementary material, etc., are defined, or are these parameters defined in the references to the models referred to a Merz FF, etc.? In that case, the authors should say so in the paper. Otherwise, it is impossible to determine what is actually being simulated.

Response & Action: Thanks a lot for your kind reminder and suggestion. According to your comments, we have summarized and added the force field (FF) parameters in Table R2 below as Supplementary Table 1 in the revised SI (page 3), including Lennard-Jones (LJ) parameters and charges for all studied ions, molecules and materials.

As you pointed out, Merz FF refers to a model (or FF) studied by Merz, etc. Supplementary Table 2 in revised SI ('table I' in your comment) further shows LJ parameters between ion and graphene, either derived with the Lorentz-Berthelot (LB) mixing rule (ϵ_{I-W}^{LB}) or taken from refs ($\epsilon_{I-W}^{ion-\pi}$).^{6,8} And the corresponding simulation results are shown in Supplementary Table 3 in revised SI ('table II' in your comment).

According to your comments, we have added these explanations in the revised main text (page 2) that, "Several different force fields (FF)⁹ are employed, such as OPLS-AA⁷, Merz^{2,3}, Netz^{4,5} and Williams⁶ FFs, the latter 3 are named by the authors who developed them. To describe ion-graphene interaction, two versions of Lennard-Jones (LJ) interaction parameters between ions and channel wall atoms (ϵ_{I-W}) were employed for each FF: one calculated by the Lorentz-Berthelot (LB) mixing rule (denoted as ϵ_{I-W}^{LB}) and the other recently optimized one which describes the ion-graphene interactions in solution accurately (denoted as $\epsilon_{I-W}^{ion-\pi}$, see Method section and Supplementary Tables 1&2 for details)". We also added them with more details to the footnotes of Supplementary Tables 1&2, e.g., 'The interaction energy (electrostatic interaction + LJ interaction) between ions and wall atoms are calculated as: $U_{I-W}(r_{I-W}) = \frac{q_{ion}q_{wall}}{4\pi\epsilon r_{I-W}} + 4\epsilon_{I-W} \left[\left(\frac{\sigma_{I-W}}{r_{I-W}} \right)^{12} - \left(\frac{\sigma_{I-W}}{r_{I-W}} \right)^6 \right] \dots$ '. The caption of Fig. 1 and Supplementary Table 2 have also been revised accordingly. For the sake of clarity, we have

denoted the LJ parameters between ion and wall atoms as ϵ_{I-W} , instead of “ $\epsilon_{ion-wall}$ ” or “ ϵ_{I-C} ” in the previous manuscript. We referred to relevant refs in the cells of these Tables, though they had been cited at the footnote of Supplementary Table 2 and elsewhere.

Table R2 | The Lennard-Jones (LJ) parameters and charges for each atom type in molecular dynamics simulations.

Atom type [a]	σ (nm)	ϵ (kJ/mol)	Charge (e)	
Merz FF ^{2,3}	Li ⁺	0.234	0.025	+1
	Na ⁺	0.261	0.122	+1
	K ⁺	0.311	0.712	+1
	Rb ⁺	0.324	0.961	+1
	Cs ⁺	0.356	1.629	+1
	Ca ²⁺	0.286	0.349	+2
	Mg ²⁺	0.249	0.062	+2
	Cl ⁻	0.385	2.224	-1
Netz FF ^{4,5}	Li ⁺	0.287	0.000615	+1
	Na ⁺	0.381	0.000615	+1
	K ⁺	0.453	0.000615	+1
	Rb ⁺	-	-	+1
	Cs ⁺	0.517	0.000615	+1
	Ca ²⁺	0.241	0.935	+2
	Mg ²⁺	0.163	0.591	+2
	Cl ⁻	0.439	0.416	-1
Williams FF ⁶	Li ⁺	0.141	1.409	+1
	Na ⁺	0.216	1.475	+1
	K ⁺	0.284	1.798	+1
	Rb ⁺	-	-	+1
	Cs ⁺	-	-	+1
	Ca ²⁺	0.241	0.94	+2
	Mg ²⁺	0.163	0.59	+2
	Cl ⁻	0.493	0.054	-1
OPLS-AA ⁷	Li ⁺	0.213	0.0765	+1
	Na ⁺	0.333	0.0116	+1
	K ⁺	0.493	0.00137	+1
	Rb ⁺	0.562	0.000715	+1
	Cs ⁺	0.672	0.000339	+1
	Ca ²⁺	0.287	0.349	+2
	Mg ²⁺	0.252	0.0624	+2
	Cl ⁻	0.442	0.493	-1
SPC/E water model ¹⁰	O	0.3166	0.65	-0.8476
	H	-	-	+0.4238
Graphene ¹¹	C	0.3214	0.49	0
hBN ¹²	B	0.33087	0.2897	+0.907
	N	0.32174	0.1979	-0.907
MoS ₂ ¹³	Mo	0.42	0.0565	+0.6
	S	0.313	1.93	-0.3
g-C ₃ N ₄ ¹⁴	C	0.343	0.44	+0.1916
	N1 [b]	0.326	0.289	-0.1883
	N2 [b]	0.326	0.289	-0.1373
	N3 [b]			

[a]: Two different LJ parameters between ion and graphene, $\epsilon_{I-W}^{ion-\pi}$ and ϵ_{I-W}^{LB} , were employed (see Supplementary Table 2). If not otherwise stated, the LJ parameters between different atoms were derived from LB mixing rule. The interaction energy (electrostatic interaction + LJ interaction) between ions and

wall atoms are calculated as: $U_{I-W}(r_{I-W}) = \frac{q_{ion}q_{wall}}{4\pi\epsilon r_{I-W}} + 4\epsilon_{I-W}[\left(\frac{\sigma_{I-W}}{r_{I-W}}\right)^{12} - \left(\frac{\sigma_{I-W}}{r_{I-W}}\right)^6]$, where r_{I-W} is the distance between ions and wall atoms; q_{ion} and q_{wall} are charges of ion and wall atom, respectively; ϵ is vacuum permittivity; ϵ_{I-W} and σ_{I-W} are LJ parameters between ions and wall atoms.
 [b]: See Supplementary Fig. 6 to distinguish N1, N2, N3 atoms of g-C₃N₄.

2. Fig. 1c seems to say that depending on whether the ion interacts with the carbon atom with a Lennard-Jones interaction with one of the two energy parameters, the distribution can be peaked on the center of the channel or displaced to sides of the channel, or am I misunderstanding what is being shown in this plot?

Response & Action: Yes, Fig. 1c is showing exactly what you pointed out. We have added such explanation to discussions on Fig. 1c in the 3rd paragraph of revised main text (page 2), “with ϵ_{I-W} increasing, the peaks of ion’s distribution profile move closer to the channel walls (Fig. 1c), which reduce $d_{ion-wall}$ ”. Thanks a lot for your helpful suggestion.

3. Plotting the residence time of hydration shell water as a function of the force-force correlation function is somewhat confusing since the physical quantity is the diffusion constant, which is proportional to the inverse of the force-force correlation function, or the friction coefficient, which is proportional to the force-force correlation function. So, perhaps it would be better to plot the residence time as a function of the diffusion constant or the friction coefficient.

Response & Action: Thanks for your helpful suggestion. According to your comments, we have revised Fig. 2d and its caption in the revised manuscript (page 6), plotting the residence time of hydration shell water as a function of the friction coefficient, as show in Fig. R4 below. In addition, we kept ΔI_{FACF} (the difference between I_{FACF} in nanochannel and that in bulk solution) as another Y axis, opposing to the $\Delta\lambda$ (the difference between the friction coefficient in nanochannel and that in bulk solution) axis in Fig. 2d, as the time integral of force-force correlation function (I_{FACF}) is just proportional to the friction coefficient.

Fig. R4 | $\Delta\lambda$ (or ΔI_{FACF}) correlates with $\tau_{channel}/\tau_{bulk}$. $\Delta\lambda$ is the difference between ion-water friction coefficient (λ) in nanochannel and that in bulk solution, which equals to ΔI_{FACF} multiplied by a constant, $\frac{1}{\gamma k_B T}$ (see Supplementary Note 1). The black curve is exponential fitting result. Shadowed area represents the prediction error.

4. In Li, et. al., *Nat. Nanotechnol.* 18, 177 (2022), which studies potassium ion flow through carbon nanotubes, the attractive interaction between the ion and the nanotube makes up for the increase of the hydrated ion's energy resulting from the removal of water molecules, which allows the ion to enter the nanotube. For the flat surfaces in these simulations, there does not seem to be any loss of hydration shell water molecules. I presume this is because for flat spaces, there is space in the directions parallel to the surfaces to accommodate the water molecules that must move out of the way to allow the hydrated ion to enter the channels?

Response: Thanks for your helpful suggestion. When ions locate in 2D nanochannels (Fig. 3, Supplementary Figs. 2-5), their solvation numbers of 1st hydration shells (HSs) are similar to the bulk solution values (only drop by ~1 sometimes when ions move quite close to the channel walls). This could be attributed to that, 'the space in the directions parallel to the surfaces accommodate the water molecules that must move out of the way to allow the hydrated ion to enter the channels', as you pointed out. In addition, 'the space in the directions parallel to the surfaces' is also referred to as 'the ring part of HS' in the manuscript.

Action: We have added the above discussions to the caption of Supplementary Fig. 2 (page 9) and Supplementary Note 5 (page 34) in the revised SI.

5. Does the friction increase because there are fewer water molecules between the ion and the surfaces?

Response: When the ion contacts closer with the water layer, the number of water molecules along channel height (z) direction between the ion and the surfaces does drop, while the water density (or the number of water molecules) in the ion's diffusion directions (parallel to the surfaces) increases, which could account for the friction increasing. Further, if we fixed an ion's z position, its diffusivity did not change even when we varied the ion-channel wall interaction parameters by a factor of ~ 100 , as Supplementary Fig. 11 and relevant discussions explain. These suggest the friction of ion-channel wall is quite smaller than that of ion-water layers for the studied 2D nanochannels with channel height (h) of ~ 1 nm. In other words, the increase of friction could be attributed to the water layers, as shown in Scheme R1.

Action: We have revised the 4th paragraph of the main text (page 4) and added Scheme R1 into the revised SI as Supplementary Scheme 1 (page 21) to emphasize the roles played by water molecules in affecting the friction.

Scheme R1 | Ions with small r_{HS} and large $d_{ion-wall}$ face less friction for diffusion (bottom), while ions with larger r_{HS} or small $d_{ion-wall}$ suffer from larger friction from water layers for diffusion (upper). The green and yellow balls stand for the 1st hydration shell for small ions and large ions, respectively. The red curves stand for the contact area (towards diffusion direction) between ion's 1st hydration and water layers. The blue area stands for water layers, with darker colors indicating higher water density.

6. Are there experimental results to compare these results to?

Response: Thanks for your helpful suggestion. The experimental diffusivity of ions in bulk solutions were well reproduced by our MD simulations (Supplementary Fig. 25). Moreover, according to your suggestion, we have run additional simulations, which yielded ion mobility, either in bulk solution or in 2D nanochannels (Fig. R5), close to the experimental results in *Nat. Nanotechnol.* **18**, 596 (2023) (ref. 15 in the main text). All these data well validate our simulation methodology. Thanks a lot for your very helpful suggestions.

Action: We have added Fig. R5 into SI as Supplementary Fig. 26 (page 46), and added the sentence “Our simulations also reproduced the experimental ion mobility (Supplementary Fig. 26), both in bulk solutions and 2D nanochannels¹⁵” in the ‘methods’ section (page 10) of the revised manuscript.

Fig. R5 | Mobilities of ions calculated by MD simulations are consistent with experimental results (*Nat. Nanotechnol.* **18, 596, 2023).** MD simulations were performed with Merz FF (ϵ_{I-W}^{LB} indicates ϵ_{I-W} derived from LB mixing rule, while $\epsilon_{I-W}^{ion-\pi}$ indicates optimized ϵ_{I-W} , see Supplementary Table 2) and salt concentration of 0.1 M in graphene nanochannel or in bulk solution.

Response to Reviewer 3

This is a very interesting simulation study where the self-diffusion and electrophoretic mobility of ions in nanochannels has been thoroughly investigated. The main strength of this paper is

that different ions corresponding to the extremes of the Hofmeister series have been compared and that mono- as well as divalent ions have been studied. Also, different ionic force fields have been compared and the results are robust with respect to a change of the force field. As a main result, a correlation between ion position and ion diffusivity has been found.

Response: Thanks for your encouragement and kind comments. We have revised our manuscript carefully according to your suggestions point-by-point, and we hope these added simulations and explanations will further validate our findings.

1. What is called Stokes-Einstein relation on page 4 is in fact the Einstein relation.

Response & Action: Thanks for your kind reminding. According to your suggestion, we have revised “Stokes-Einstein relation” to “Einstein relation” in the revised manuscript (page 4) and highlighted the text in yellow.

2. The friction coefficient is extracted using the integral over the force-autocorrelation function, which only gives a converging result in the limit of an immobilized ion (or an ion with infinite mass), as correctly pointed out by the authors in the Supplement. However, the friction coefficient of an object is modified by confinement, as was recently shown (“External Potential Modifies Friction of Molecular Solutes in Water” Daldrop et al, PHYSICAL REVIEW X 7, 041065 (2017)). This effect is due to the effect that the water exchange dynamics around an object depends on whether the object is free to move in response to the hydration water dynamics or not. In fact, the ion friction coefficient has been shown to increase due to confinement by about 5% for a negative ion and by about 20% for a positive ion (“Memory-kernel extraction for different molecular solutes in solvents of varying viscosity in confinement” Kowalik et al, PHYSICAL REVIEW E 100, 012126 (2019)). The authors should have observed these modifications in their simulations, a discussion of this is in order. The alternative would be to obtain the friction from the memory kernel, which however is rather involved to extract. I do not believe that the mechanism for the friction changes due to this confinement effect, but this should be pointed out clearly in the paper.

Response: Thanks a lot for your very helpful suggestions. Yes, we did observe that, ion diffusivity calculated from force-autocorrelation function (D^{FACF}) could be lower than those from Mean Square Displacement (D^{MSD}) by up to 20% in Supplementary Fig. 12 as you pointed out. According to your comments, we further calculated $D_{channel}^{FACF}/D_{bulk}^{FACF}$ for ions fixed at

different positions in the nanochannel, where $D_{channel}^{FACF}/D_{bulk}^{FACF} \sim d_{ion-wall}$ shows similar rule like $D_{channel}^{MSD}/D_{bulk}^{MSD} \sim d_{ion-wall}$ (Fig. R6, D^{FACF} and D^{MSD} denote diffusivity calculated with the FACF or MSD, respectively). The reason may be attributed to that, when we calculated $D_{channel}^{FACF}/D_{bulk}^{FACF}$, the fixing position (confinement) effect in nanochannel cancel with that in bulk solution to a large extent. This indicates the confinement effect does not affect ‘the mechanism for the friction changes’, exactly as you pointed out. It also further confirm the broad applicability of our findings.

Action: We have added Fig. R6 as Supplementary Fig. 13 in the revised SI (Page 23) with the above discussions added into its caption. We have also added more discussion and above refs to the revised manuscript (page 4, the 1st paragraph of ‘Correlations of physics quantities’ section) and pointed out the confinement effect in the caption of Supplementary Fig. 12 (Page 22).

Fig. R6 | $D_{channel}^{FACF}/D_{bulk}^{FACF} \sim d_{ion-wall}$ and $D_{channel}^{MSD}/D_{bulk}^{MSD} \sim d_{ion-wall}$ follow similar rule. The shadowed area is identical with that in Fig. 1d. $D_{channel}^{FACF}$ and D_{bulk}^{FACF} are diffusivities calculated from force-autocorrelation function simulations which fixed the ion at different positions in the nanochannel or in bulk solution respectively. The data of $D_{channel}^{MSD}/D_{bulk}^{MSD}$ (the superscript MSD indicates the diffusivity data are calculated from the MSD) are taken from Fig. 1d. Although $D_{channel}^{FACF}$ sometimes deviates from D_{bulk}^{FACF} (Supplementary Fig. 12), $D_{channel}^{FACF}/D_{bulk}^{FACF} \sim d_{ion-wall}$ and $D_{channel}^{MSD}/D_{bulk}^{MSD} \sim d_{ion-wall}$ follow similar rule, as the fixing position (confinement) effect in nanochannel cancels with that in bulk solution to a large extent.

References

1. Liao, S., Ke, Q., Wei, Y. & Li, L. Water's motions in x-y and z directions of 2D nanochannels: entirely different but tightly coupled. *Nano Res.* **16**, 6298-6307 (2023)
2. Li, P. F., Roberts, B. P., Chakravorty, D. K. & Merz, K. M. Rational Design of Particle Mesh Ewald Compatible Lennard-Jones Parameters for +2 Metal Cations in Explicit Solvent. *J. Chem. Theory Comput.* **9**, 2733-2748 (2013)
3. Li, P. F., Song, L. F. & Merz, K. M. Systematic Parameterization of Monovalent Ions Employing the Nonbonded Model. *J. Chem. Theory Comput.* **11**, 1645-1657 (2015)
4. Horinek, D., Mamatkulov, S. I. & Netz, R. R. Rational design of ion force fields based on thermodynamic solvation properties. *J. Chem. Phys.* **130**, 124507 (2009)
5. Mamatkulov, S., Fyta, M. & Netz, R. R. Force fields for divalent cations based on single-ion and ion-pair properties. *J. Chem. Phys.* **138**, 024505 (2013)
6. Williams, C. D., Dix, J., Troisi, A. & Carbone, P. Effective Polarization in Pairwise Potentials at the Graphene-Electrolyte Interface. *J. Phys. Chem. Lett.* **8**, 703-708 (2017)
7. Jorgensen, W. L., Maxwell, D. S. & Tirado-Rives, J. Development and testing of the OPLS all-atom force field on conformational energetics and properties of organic liquids. *J. Am. Chem. Soc.* **118**, 11225-11236 (1996)
8. Liao, S. *et al.* Molecular Dynamics Simulation of Ion Adsorption at Water/Graphene Interface: Force Field Parameter Optimization and Adsorption Mechanism. *Chem. J. Chinese Universities* **44**, 184-195 (2023)
9. Li, P. & Merz, K. M., Jr. Metal Ion Modeling Using Classical Mechanics. *Chem. Rev.* **117**, 1564-1686 (2017)
10. Berendsen, H. J. C., Grigera, J. R. & Straatsma, T. P. The missing term in effective pair potentials. *J. Phys. Chem.* **91**, 6269-6271 (1987)
11. Werder, T., Walther, J. H., Jaffe, R. L., Halicioglu, T. & Koumoutsakos, P. On the water-carbon interaction for use in molecular dynamics simulations of graphite and carbon nanotubes. *J. Phys. Chem. B* **107**, 1345-1352 (2003)
12. Rajan, A. G., Strano, M. S. & Blankschtein, D. Ab Initio Molecular Dynamics and Lattice Dynamics-Based Force Field for Modeling Hexagonal Boron Nitride in Mechanical and Interfacial Applications. *J. Phys. Chem. Lett.* **9**, 1584-1591 (2018)
13. Chen, H. *et al.* Protein Translocation through a MoS₂ Nanopore: A Molecular Dynamics Study. *J. Phys. Chem. C* **122**, 2070-2080 (2018)
14. Wang, Y. *et al.* Water Transport with Ultralow Friction through Partially Exfoliated g-C₃N₄ Nanosheet Membranes with Self-Supporting Spacers. *Angew. Chem. Int. Ed.* **56**, 8974-8980 (2017)
15. Goutham, S. *et al.* Beyond steric selectivity of ions using angstrom-scale capillaries. *Nat. Nanotechnol.* **18**, 596-601 (2023)

----- The end -----

Prof. Haihui Wang
Changjiang Chair Professor
NSFC outstanding Young Investigator
Fellow of The Royal Society of Chemistry
Department of Chemical Engineering
Tsinghua University
Beijing 100084, China
Tel.: +86-10-62793144
E-mail: cehhwang@tsinghua.edu.cn

Response to the Reviewers' Comments

Many thanks to the reviewers for their valuable comments and suggestions. The followings are the point-by-point answers to the comments:

Response to Reviewer 1

Reviewer #1: In this revised version, the authors have addressed all my concerns and the manuscript has been improved significantly. Now I can recommend its publication.

Response: Thank you very much for your positive evaluation of our manuscript. Your support motivates us to continue our research in this field.

Response to Reviewer 2

Reviewer #2: The authors appear to have answered all of my questions. As far as I am concerned, you should publish this manuscript.

Response: Thank you very much for your positive evaluation of our manuscript. Your support motivates us to continue our research in this field.

Response to Reviewer 3

Reviewer #3: I carefully read the reply letter and the revised paper and conclude that the authors have adequately addressed my comments and also the comments by other referees. I therefore recommend publication of the paper as is.

Response: Thank you very much for your positive evaluation of our manuscript. Your support motivates us to continue our research in this field.

----- The end -----